# We’re Not Gonna Fall: Depressive Complaints, Personal Resilience, Team Social Climate, and Worries about Infections among Hospital Workers during a Pandemic

**DOI:** 10.3390/ijerph18094701

**Published:** 2021-04-28

**Authors:** Bram P. I. Fleuren, Lieze T. Poesen, Rachel E. Gifford, Fred R. H. Zijlstra, Dirk Ruwaard, Frank C. van de Baan, Daan D. Westra

**Affiliations:** 1Department of Work and Social Psychology, Faculty of Psychology and Neuroscience, Maastricht University, 6229 ER Maastricht, The Netherlands; l.poesen@maastrichtuniversity.nl (L.T.P.); fred.zijlstra@maastrichtuniversity.nl (F.R.H.Z.); 2Department of Health Services Research, Faculty of Health, Care and Public Health Research Institute (CAPHRI), Medicine and Life Sciences, Maastricht University, 6229 GT Maastricht, The Netherlands; r.gifford@maastrichtuniversity.nl (R.E.G.); d.ruwaard@maastrichtuniversity.nl (D.R.); f.vandebaan@maastrichtuniversity.nl (F.C.v.d.B.); d.westra@maastrichtuniversity.nl (D.D.W.)

**Keywords:** depressive complaints, personal resilience, team social climate, worries about infections, COR-theory, hospital workers, COVID-19 pandemic, COVID-19

## Abstract

Maintaining hospital workers’ psychological health is essential for hospitals’ capacities to sustain organizational functioning during the COVID-19 pandemic. Workers’ personal resilience can be an important factor in preserving psychological health, but how this exactly works in high stakes situations, such as the COVID-19 pandemic, requires further exploration. Similarly, the role of team social climate as contributor to individual psychological health seems obvious, but how it exactly prevents workers from developing depressive complaints in prolonged crises remains under investigated. The present paper therefore applies conservation of resources theory to study the relationships between resilience, team social climate, and depressive complaints, specifically focusing on worries about infections as an important explanatory mechanism. Based on questionnaire data of 1126 workers from five hospitals in the Netherlands during the second peak of the pandemic, this paper estimates a moderated-mediation model. This model shows that personal resilience negatively relates to depressive complaints (β = −0.99, *p* < 0.001, 95%CI = −1.45–−0.53), partially as personal resilience is negatively associated with worries about infections (β = −0.42, *p* < 0.001, 95%CI = −0.50–−0.33) which in turn are positively related to depressive complaints (β = 0.75, *p* < 0.001, 95% CI = 0.31–1.19). Additionally, team social climate is associated with a lower effect of worries about being infected and infecting others on depressive complaints (β = −0.88, *p* = 0.03, 95% CI = −1.68–−0.09). These findings suggest that resilience can be an important individual level resource in preventing depressive complaints. Moreover, the findings imply that hospitals have an important responsibility to maintain a good team social climate to shield workers from infection related worries building up to depressive complaints.

## 1. Introduction

The global COVID-19 pandemic relentlessly confronts hospitals and hospital workers with complex and unprecedented problems [1,2,3,4,5], threatening hospital workers’ psychological health severely [6,7,8,9,10,11,12]. Hospitals must be resilient in navigating these problems [13,14,15] and their personnel–as hospitals’ primary resource [16,17,18]–are essential in achieving such resilience [19]. Indeed, securing sufficient healthcare workers in terms of both quantity and quality is paramount for public health [14,20]. However, the immense threats to hospital workers’ psychological health that flow from the COVID-19 pandemic complicate the situation [6,7,8,9,10,11,12]. Depression, anxiety, insomnia, distress, and burnout show vastly increased prevalence among healthcare workers during the pandemic [9,21]. These negative outcomes are associated with many factors, such as lacking personal protective equipment [22,23], close contact with COVID-19 patients [24,25], longer working hours [26], worries about their family [27], and fear of infection [28,29]. Given the need to sustain the functioning of the healthcare workforce [30] amidst these many sources of strain, understanding how to protect hospital workers’ health and well-being is essential.

Research so far suggests a number of factors that can protect hospital workers’ health and well-being. A first important individual protective factor is personal resilience [31,32,33]. Personal resilience is the ability of a person to bounce back or recover from stress [34]. Earlier research reveals that resilience can safeguard individuals against mental illness and depression [32,35]. However, the mechanisms by which resilience protects hospital workers in a prolonged crisis are not known. Second, a potential organizational/team level protective factor is the support workers receive from their team [36,37]. Relatedly, Rangachari et al. [14] recommend in this special issue that a positive work environment of trust, psychological safety, and empowerment can protect workers and foster organizational resilience. However, how this works exactly and to what extent a team social climate can protect individuals’ psychological health–particularly during the COVID-19 pandemic–still needs to be studied empirically [6,9,10,11]. Third, studies that do consider the protective and risk factors regarding the individual psychological health of healthcare workers during the pandemic typically only target frontline staff (medical and nursing staff). Consequently, little is known about these effects on the full range of hospital workers [9], who are likely to be also majorly affected. Similarly, most existing studies [22,23,24,25,26,27,28,29] on this topic target the first peak of COVID-19 infections, and do not extend to potential enduring effects.

Given the urgent need for insights into how hospital workers’ psychological health can be preserved and how personal resilience and team climate contribute to this, the present paper provides a large-scale quantitative study on the relationships between these variables. Specifically, this study uses Conservation of Resources (COR) theory [38,39] to suggest that personal resilience serves as a resource that protects hospital workers against depressive complaints. Subsequently, this study explores worries about infections as a common threat that many hospital workers face, via which hospital workers lower in resilience can develop depressive complaints. Additionally, team social climate is positioned as a contextual resource that may protect hospital workers from such worries leading to depressive complaints. By studying these relationships in a large and diverse sample of hospital workers, this paper generates important and generalizable novel insights. That is, this paper establishes the relevance of, and the mechanisms (i.e., lower levels of worrying) by which personal resilience and the team context can prevent workers from developing depressive complaints. These insights offer theoretical contributions by testing COR theory and demonstrating a clear example of how one resource’s effects can overrule the effects of another [39,40], but also enabling hospitals to protect their workers in practice. Hereby, the study builds an important bridge between theoretical discussions on organizational factors and personal resilience and workers’ psychological health in practice.

### Hypotheses

The present study draws on Conservation of Resources (COR) theory to identify mechanisms that prevent hospital workers from developing depressive complaints amidst crises. COR theory suggests that individuals are motivated to protect and acquire resources. Resources are entities that individuals value and that typically contribute to the individuals’ capacity to achieve goals [39]. In that sense, individuals’ psychological health (i.e., in this study, low depressive complaints) generally constitutes an important resource in individuals’ pursuit for a good life and happiness [41]. COR theory also posits that resources can be threatened and threats to resources may elicit resource protective tendencies in individuals [40]. Moreover, according to COR theory, such resource protective tendencies require the investment of resources, such that individuals with more resources are better equipped to handle threats. Therefore, following COR theory, hospital workers who face strenuous circumstances that threaten their psychological health by eliciting depressive complaints will strive to protect it, and those workers with the resources to do so will succeed better at doing so.

Personal resilience can be positioned as an important individual level resource. That is, personal resilience refers to individuals’ capacity to handle difficult circumstances by bouncing back when facing adversity [42]. Individuals higher in personal resilience will therefore be better capable of handling threats to their psychological health [32,33]. As such, for our first hypothesis (Hypothesis 1) is that personal resilience is expected to negatively relate to depressive complaints (i.e., as a negative indicator for psychological health). If this hypothesis is correct, hospitals can arguably prevent workers from developing depressive complaints by facilitating the development of their resilience [43].

To understand exactly how resilience functions as a resource, it is crucial to consider mechanisms by which it facilitates individuals in maintaining psychological health. In the context of an infectious disease, hospital workers constantly face a threat of being infected and infecting others. This arguably elicits a need for control over not getting infected or infecting relevant others, while it is hard to exert this control because of the high infectiousness and uncertainty regarding one’s own infection status [29]. Consequently, individuals are likely to experience worry in their attempts to exert cognitive control over the situation [44,45,46]. In line with previous studies documenting relationships between worries about infection and psychological distress among healthcare workers [47,48], such worries are likely positively associated with depressive complaints (Hypothesis 2a). Moreover, as personal resilience is likely to help individuals in perceiving the situation as less threatening [49,50,51], individuals high in resilience will be likely to worry less (Hypothesis 2b). Taken together, these two predictions imply a mediation effect; individuals higher in personal resilience will worry less and thus are likely to experience fewer depressive complaints (Hypothesis 2c).

In terms of contextual resources, a good team social climate could help hospital workers in avoiding depressive complaints. Team climates can have several foci [52], but important elements for a team social climate are trust, good communication, cohesion and good relationships among team members [53]. It is known that individuals generally value inclusion in groups and that being part of a group that is held in positive esteem is associated with well-being [54]. As such, team social climate is hypothesized to be negatively associated with depressive complaints (Hypothesis 3a). More importantly, membership in groups that have a team social climate may provide individuals with access to indirect social support [55]. Teams at work constitute particularly relevant groups, because they offer individuals comparable perspectives and thus validation of thoughts and feelings [56,57], which could reduce the effect that worries have on depressive complaints. Indeed, studies show that social support helps individuals cope better with stress [58,59], prevents moral injury [60] and burnout [61,62], and buffers COVID-19 worries’ negative effect on health [63]. Therefore, team social climate is expected to function as an important contextual resource in preserving psychological health, by buffering the effect of worries about infections on depressive complaints (Hypothesis 3b).

Recent work on COR theory has suggested that some resources have more value to individuals than others as they are more important for goal attainment [40]. This notion is referred to as the substitutive effect of resources. In light of the aforementioned hypotheses, the value of personal resilience in preventing depressive complaints by reducing worrying could be exceeded by team social climate. That is, if personal resilience’s effect on depressive complaints indeed primarily works via worry reduction, and team social climate buffers the effect of worries about infections on depressive complaints, team social climate as a contextual resource could meaningfully substitute for personal resilience as an individual level resource. In sum, such a substitutive effect of team social climate, would suggest that the effect of personal resources on depressive complaints as mediated by worries about infections is moderated by team social climate (Hypothesis 4).

Given the several possibilities for modeling a set of variables, it is essential to specify the theoretical rationale for the position of each variable in the research model [64] that follows from the hypotheses (Figure 1). First, as this paper aims to identify how hospital workers can be protected during crises, depressive complaints are the most logical dependent variable of interest. Second, as personal resilience is a trait-like individual characteristic, it should logically be an initial predictor and cannot function as a mediator variable in this non-interventional study. Third, as worries about infections are a state-like response to strenuous conditions among hospital workers that is likely to be more present in workers with lower resilience, these worries make most sense as mediator variable. Indeed, traits like personal resilience can only link to outcomes via the situation-specific responses they elicit, such that worries about infections can be a situation specific utterance of lower personal resilience scores in the pandemic. Fourth, team social climate is most likely to function as a protective resource given a perceived situation specific threat, such that it makes most sense as a moderator of the relationship between worries about infections and depressive complaints. It is thus of lesser interest to consider its moderating effect on the relationships between personal resilience and worries about infections and depressive complaints, respectively.

## 2. Materials and Methods

### 2.1. Study Design

Data for this study were collected as part of an ongoing large-scale observational longitudinal study among five hospitals in the Netherlands. This study received approval from the Ethics Review Committee Psychology and Neuroscience of Maastricht University (protocol code ERCPN-230_130_11_2020), started in December 2020 and will conclude in September 2021. Specifically, the data analyzed in this paper were collected in December 2020 as part of the first wave of this large-scale longitudinal study. At this time, the Netherlands were at the height of the second peak of COVID-19 infections [65].

The recruitment approach consisted of several steps. First, the boards of the five hospitals participating in this study were contacted, who each appointed a liaison officer in their hospital. These five liaison officers provided us with e-mail addresses of all the employees of their hospital (*n* = 23,306) and advertised the diary study in the hospital staff newsletter and on their internal website (intranet). All employees (i.e., healthcare professionals as well as any other non-clinical hospital workers) of all five hospitals were then invited to voluntarily participate in the study via e-mail at the beginning of December 2020. This e-mail included a brief description of the purpose of the study on the basis of which the hospital workers could choose to proceed to the survey.

Those who proceeded to the survey received full information about the study purposes and made an informed choice regarding participation. That is, hospital workers were directed to an information webpage of the study where the full purpose of the study, the usage of personal data, and privacy policies were stated. Those wishing to participate in the study clicked on a link to proceed to the informed consent page and registered with an e-mail address of choice. After giving consent, participants could first participate in a general survey that captured several demographic variables, personality traits, and experiences regarding working during the first peak of COVID-19 infections in the Netherlands. Subsequently, participants were directed to another questionnaire primarily aimed at capturing various work characteristics, including COVID-19 related working conditions. This part of the questionnaire was open for the first week of the study. In the second week, participants could participate in short daily questionnaires for the one-week period to capture daily experiences. Finally, in the third week of the study, participants received a questionnaire targeting variables related to work functioning at the individual level. As this approach was planned to be repeated three times in the future with three-month intervals, participants were thanked, received a message alerting them of the planned data collection throughout 2021, and were given the opportunity to participate in a raffle to win one of ten vouchers of €50.00.

All questionnaires were administered via the online survey platform Qualtrics^®^ and repeated measures were connected via the SOTO software platform. Data from participants were stored in full accordance with GDPR guidelines and European legislation for data protection. That is, participation in the study was fully anonymous, all responses were treated with full confidentiality, and were stored on protected servers. 

### 2.2. Study Variables

Depressive complaints were measured in the third week of the study using a Dutch translation of the validated PHQ-9 (Patient Health Questionnaire) [66,67]. PHQ-9 is a self-administered questionnaire to screen for depression, by checking for the severity of symptoms during the past two weeks. The nine items were scored on a four-point Likert type scale, ranging from 0 to 3 (“Not at all” to “Nearly every day”). The total score, ranging from 0 to 27, was calculated for each participant and employed as a continuous variable in the statistical analysis. Given the potentially emotionally laden nature of the questions in the PHQ-9, links to the national suicide hotline and the hospitals’ psychosocial team were provided with these items.

Personal resilience was measured in the general survey at the beginning of the study. Specifically, a Dutch translation of three items of the validated Brief Resilience Scale [34,68] was used. This scale was specifically designed to measure resilience when dealing with ongoing health-related stresses [34]. The three selected resilience items (e.g., “I have a hard time making it through stressful events.”) were rated on a five-point Likert type scale, ranging from 1 to 5 (“Completely disagree” to “Completely agree”).

Worries about infections was also measured in the first week of the study. For this purpose, three items taken from the HEROES study [69] were adapted to reduce the complexity of formulations. Specifically, the respondents were asked to rate on a five-point type Likert type scale, ranging from 1 to 5 (“Not worried at all” to “Very worried”) how worried they have been in the past three months about: (a) being infected themselves with COVID-19; (b) infecting patients/colleagues with COVID-19; and (c) infecting family/loved ones with COVID-19. These three items were then combined into an average ‘worries about infections’ score.

Team social climate was also measured as part of the first week questionnaire. The construct was based on four items [70,71,72] that were scored on a five-point Likert type scale, ranging from 1 to 5 (“Completely disagree” to “Completely agree”). The specific items were “We are a cohesive team”, “In my team, people can completely trust each other”, “Relevant information is openly shared with all team members”, and “Interpersonal relationships in my team are excellent”. These items were averaged together to create the overall team social climate score, such that a higher mean score on the variable represented a better team social climate. 

### 2.3. Statistical Analysis

All data analyses were conducted with IBM^®^ SPSS^®^ Statistics 26.0 software. First, descriptive analyses were performed to explore frequencies, means, and standard deviations, depending on the type of the variable. Second, Cronbach’s alphas for all scales included in the study as well as zero-order correlations among all study variables were estimated. Third, the full moderated mediation model was estimated using the PROCESS 3.4 plug-in for IBM^®^ SPSS^®^ Statistics 26.0 (IBM Corp., Armonk, NY, USA), model 14 [73]. This model includes several consecutively estimated effects. That is, the model first estimated the direct effect of resilience as independent variable on worries about infections as mediator. Second, the model simultaneously estimated the direct effects of resilience (independent variable), worries about infections (mediator), team social climate (moderator), and the interaction between worries about infections and team social climate (interaction) on depressive complaints as dependent variable. Third, the model estimated the direct effect and the conditional (i.e., depending on team social climate) indirect (i.e., via worries about infections) effect of resilience on depressive complaints as dependent variable. To determine if indirect effects are significantly different from zero, bootstrapping was performed to estimate standard errors and confidence intervals. The number of bootstrapping procedures to be performed was fixed to a maximum of 50,000 iterations to achieve optimally robust results. Moreover, a correction for potential heteroscedasticity was applied by specifying Cribrari-Neto estimation and the continuous variables in the interaction effects were mean-centered (i.e., team social climate and worries about infections). Finally, simple effects for the interaction effect at the mean and one standard deviation below and above the mean of worries about infections were specified.

Several supplementary post-hoc analyses were performed to estimate mediation effect sizes, to address nesting in occupational groups, to correct for potential confounding of working hours, and to check for alternative moderating effects of team social climate. The results of these analyses are briefly discussed in the results section. However, as the results of these analyses were inconsequential for the main analyses above, details are not included in the paper.

## 3. Results

### 3.1. Participants

1126 hospital workers participated in the first wave of the study in the first survey. Table A1 in Appendix A depicts the demographic and employment characteristics for all of the participants. As respondents were not forced to answer all the questions, missing values for the descriptive variables are also reported in the Appendix A Table A1. Moreover, respondents could drop out during the consecutive measurement occasions of the first wave of the study. Hence, for subsequent correlational analyses, numbers of participants are reported for each statistic.

### 3.2. Descriptives for the Main Study Variables

Before turning to the main model of interest, descriptive analyses were conducted. Table 1 shows an overview of correlations, Cronbach’s alphas, means, standard deviations and ranges for each of the study variables of interest. As can be observed from the table, all study variables correlate in the directions suggested by Hypotheses 1, 2a,b, and 3a.

### 3.3. Moderated Mediation Model

The moderated mediation model estimated effects in several consecutive steps. Table 2 reports the results for each step of the analysis. At the first step, the direct effect of personal resilience as independent variable on worries about infections as mediator was negative and significantly different from zero (β = −0.42, *p* < 0.001, 95% CI = −0.50–−0.33). The explained variance (R^2^ = 0.12, *p* < 0.001) for this first model differed significantly from zero. Second, the concurrently estimated direct effects of personal resilience (β = −0.99, *p* < 0.001, 95% CI = −1.45–−0.53), worries about infections (β = 0.75, *p* < 0.001, 95% CI = 0.31–1.19), and team social climate (β = −0.52, *p* = 0.04, 95% CI = −1.02–−0.02) also differed significantly from zero. Additionally, the interaction effect between worries about infections and team social climate (β = −0.88, *p* = 0.03, 95% CI = −1.68–−0.09) in this second model also differed significantly from zero (Table 2, Figure 1). The variance explained by this model differed significantly from zero (R^2^ = 0.14, *p* < 0.001) and the increase compared to the previous model (R^2^-change = 0.02, *p* = 0.03) also differed significantly from zero. The results in Table 2 align exactly with Hypotheses 1 to 3b. At the third step of the analysis, the full moderated–mediation effect (i.e., the effect of personal resilience via worries about infections moderated by team social climate) on depressive complaints was estimated via bootstrapping. This final step revealed that the overall moderated mediation effect on depressive complaints also differed significantly from zero (β = 0.37, 95% CI = 0.07–0.73), confirming the full hypothesized moderated mediation effect (Hypothesis 4).

Finally, simple slope analyses were probed at the mean value, −1SD and +1SD of team social climate, to explore how the effect of personal resilience as mediated by worries about infection on depressive complaints differed per level of team social climate. For the mean value (β = −0.31, 95% CI = −0.51–−0.13) and −1SD (β = −0.58, 95% CI = −0.93–−0.27) below the mean value of team social climate this effect was negative and significantly different from zero. However, for +1SD above the mean value of team social climate (β = −0.05, 95% CI = −0.31–0.21) the effect no longer differed significantly from zero (Figure 2).

An additional post-hoc analysis estimated the effect sizes of the direct and indirect effects in a simple mediation model. Specifically, this model included the direct effect of personal resilience on depressive complaints as well as the indirect effect of personal resilience via worries about infections on depressive complaints in PROCESS model 4 [73]. Effect size estimates revealed that the total effect size of the effect of personal resilience is −0.29, which can be partitioned into a direct effect size of −0.23 and an indirect (i.e., via worries about infections) effect size of −0.06. As all effects differ significantly from zero, worries about infections only partially mediate the effect of personal resilience on depressive complaints, by accounting for 21% of the total.

Further post-hoc analyses were performed to check for variance due to nesting in occupational groups, testing alternative moderations of team social climate, and to check for working hours as potential covariate. First, a series of multilevel regression analyses revealed that nesting in occupational groups was not associated with any significant intercept variation or random effects for the relationships tested in our model. Second, alternatively specified moderated–mediation models revealed that the moderating effect of team social climate on the relationship between worries about infections and depressive complaints proved to be the most relevant. Finally, including working hours in addition to the moderated–mediation model reported in Table 2 showed that working hours did not have a significant confounding effect. For conciseness, results from these analyses are not included in the present paper, but output is available from the corresponding author.

## 4. Discussion

This study aims to examine the relationship between personal resilience and hospital workers’ depressive complaints and potential mechanisms involved. The study shows that personal resilience is indeed negatively associated with depressive complaints among hospital workers during the second peak of the COVID-19 pandemic in the Netherlands. Worries about infections seem to be an important mechanism in this relationship, as personal resilience is associated with fewer worries, which are in turn associated with fewer depressive complaints. Importantly, team social climate moderates the connection between worries about infections and depressive complaints, such that hospital workers’ worries about infections are less strongly associated with depressive complaints if they have a good team social climate. In fact, confirmation is found for the full moderated-mediation model (i.e., the effect of personal resilience via worries about infections on depressive complaints is moderated by team social climate). From a Conservation of Resources (COR) theory perspective [38,39], these findings suggest that personal resilience and team social climate are relevant resources that hospital workers can benefit from during crises. The remainder of this section discusses each of the findings and their implications in more detail, considers potential limitations of this study, and ends with a concluding paragraph.

This study firstly identifies personal resilience as an important individual level resource in preventing depressive complaints. This finding confirms Hypothesis 1 and aligns with previous studies that connect personal resilience to fewer psychological problems [32,35]. Importantly, by showing a negative association with worries about infections as explanatory mechanism (Hypotheses 2a–c) in the relationship between personal resilience and depressive complaints, this study advances insights on how personal resilience works. This provides further evidence for COR theory’s predictions regarding the role of resources as protective factors for psychological health. By doing so in a diverse sample of hospital workers as recommended by de Kock et al. [9], the present paper offers hospitals generalizable directions for preventing depressive complaints during crises. That is, as personal resilience is considered a trainable attribute of individuals, these insights can be used to maintain hospital workers’ functioning during crises [43]. Hospitals can for example implement mindfulness-based resilience training [74,75] to boost personal resilience and thereby reduce hospital workers worries and depressive complaints.

The role of worries about infections in producing depressive complaints is another important finding of this study, beyond its implications for personal resilience. Namely, when hospital workers have to work during an infectious disease pandemic, such worries are likely to occur at some point, even among the more resilient workers. As worries are predictive of depressive complaints, they can also be targeted directly. Hospitals can, for example, achieve this by providing adequate personal protective equipment [23], testing opportunities, and the fastest access to vaccines, in this case against COVID-19. As the potential connection between worrying about infections and reduced mental health is known [47,48], perhaps these findings may underscore the relevance of prioritizing hospital staff in vaccination programs during pandemics specifically.

The findings regarding team social climate emphasize the relevance of contextual resources in maintaining hospital workers’ functioning. That is, team social climate is identified as an important resource that is associated negatively and directly with depressive complaints (Hypothesis 3a). This direct effect of team social climate echoes previous findings of the importance of relevant groups for well-being [54]. Hereby, this study empirically shows that hospitals should foster team climate, especially during crises [14]. Moreover, team social climate is found to buffer the direct and mediating (i.e., from personal resilience to depressive complaints) effects of worries about infections (Hypothesis 3b and 4 respectively) on depressive complaints. This finding connects to previous research on the buffering effects of social support for mental health outcomes [58,59,76]. Importantly, it suggests that, even when workers do not have high levels of personal resilience, their psychological health can be protected by fostering a team social climate. Hereby, this study demonstrates a substitutive effect of resources (i.e., team social climate can replace or compensate for personal resilience to some extent) [39]. As such, hospitals must preserve the cohesion, trust, communication and interpersonal relations in teams to maintain their workers’ functioning, especially during crises. Particularly because individual workers cannot exert full influence over the team climate, these findings stress an important organizational responsibility for sustaining hospital workers’ functioning [30].

### Potential Limitations

A first potential limitation of this study is its exclusive reliance on self-report measures. This is a commonly cited concern in research among employees, but the extent to which it is actually problematic and applies to a specific study must be carefully considered per study [77,78]. Reliance on self-report measures exclusively might be problematic because it can–but does not necessarily have to–inflate correlations due to common method variance [78]. This is particularly important when study variables are all measured at the exact same time point [79]. As the main outcome variable (i.e., depressive complaints) is measured in the third week of the study and all the predictors of interest are measured in the first week, the impact of common method bias seems to be conceptually restricted to relationships among predictor variables. Here it is important to note that personal resilience (as trait) and team social climate (as perceived team characteristic) are less time-variant than worries about infections. Therefore, the mediation model as tested makes sense conceptually despite possible considerations of common method bias. To circumvent this issue, future studies should ideally measure all of the variables at different time points.

A second potential limitation of this study is that team social climate is measured by individual perceptions. Ideally, a climate measure contains a team-level weighted aggregate score of such perceptions [52]. However, due to the limited number of individual participants per team in our sample, this was not possible to include as a meaningful variable. As such, future research might strive to replicate our findings with climate perceptions of the complete set of members of a team. Importantly, this limitation does not invalidate our results, but interpretations are primarily restricted to individual perceptions of team social climate. However, this limitation only has a minor impact on the interpretation of our results, as the way individuals personally perceive their team’s social climate is most central to their potential depressive complaints.

Third, some readers might consider the minimalist approach to including covariates in the study models a potential drawback. However, this study deliberately did not include ‘usual-suspect’ covariates (e.g., gender, education level, and age) in its analysis, because the inclusion of covariates must be based on theoretical considerations. As there were no theoretical reasons to assume that, for example, such demographic factors would bias our outcomes when not included, recommendations from the literature not to include them a priori are followed [80].

Lastly, our study might suffer from two similar potential selection effects. That is, it is possible that hospital workers who face particularly high levels of work pressure would not participate in our study. Relatedly, our sample might be biased through a ‘healthy worker effect’ [81] meaning that the most severe cases of worrying or depressive complaints are not included in the sample, as they were not working during the study period. Importantly, no specific systematic basis (e.g., specific personal reasons) for non-participation is clearly present in the data. Nonetheless, although these potential selection effects do not invalidate our results directly, they might restrict the generalizability of findings to workers with milder complaints. Arguably, this particularly applies to the descriptive results (e.g., means and standard deviations) and to a lesser extent to the correlational findings. Future research should strive to include these hard-to-reach groups of workers for full generalizability.

## 5. Conclusions

The present paper demonstrates the relevance of personal resilience and team social climate in preventing depressive complaints among hospital workers during a pandemic. Personal resilience as an individual level resource can reduce worries and may thereby reduce depressive complaints. Team social climate as a contextual resource is also associated with fewer depressive complaints, but also seems to reduce the association between worries about infections and depressive complaints. Hereby, this paper offers hospitals useful directions for preserving the psychological health of all types of hospital workers. Namely, hospitals can arguably train their workers to become more resilient, reduce worries about infections (e.g., by testing, protecting, and vaccinating), and foster a team social climate to prevent depressive complaints. As such, this paper offers an important validation of conservation of resources theory that emphasizes the important responsibility of hospitals to act and protect their workers during pandemics. After all, the resilience of an organization depends largely on the functioning of its workforce.

## Figures and Tables

**Figure 1 ijerph-18-04701-f001:**
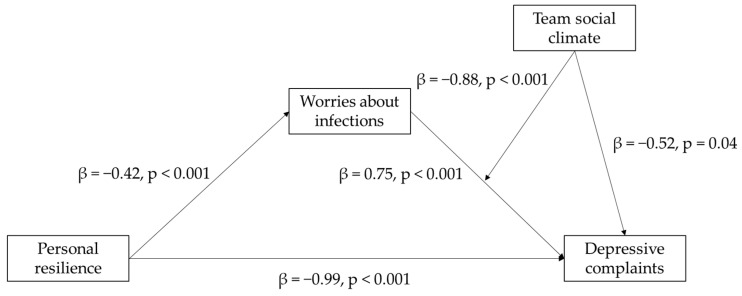
Moderated mediation model for the effect of personal resilience via worries about infections as moderated by team social climate.

**Figure 2 ijerph-18-04701-f002:**
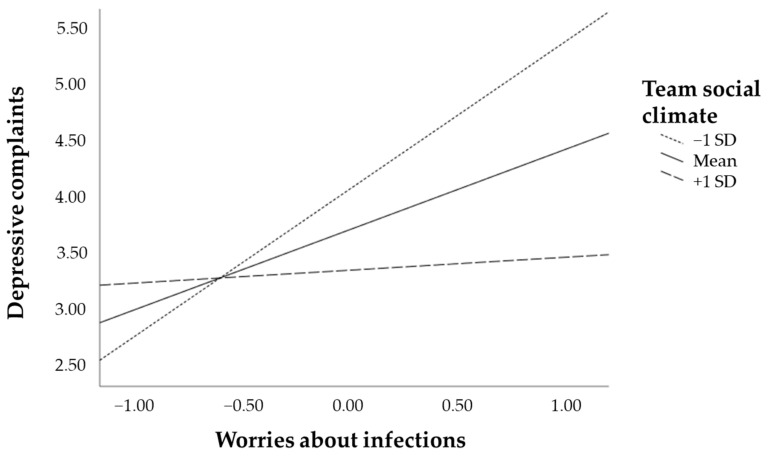
Simple slopes plot for mediated personal resilience on depressive complaints, via worries about infections for the mean, one standard deviation above, and one standard deviation below the mean of team social climate.

**Table 1 ijerph-18-04701-t001:** Correlations, Cronbach’s alphas, means, standard deviations, and ranges for the main study variables.

	1	2	3	4	Mean	*n*	SD	Range
1. Personal resilience	0.72	-	-	-	3.67	1068	0.70	1–5
2. Worrying about infections	−0.29 ^a^	0.84	-	-	2.99	1034	0.86	1–5
3. Team social climate	0.18 ^a^	−0.08 ^b^	0.87	-	3.73	1025	0.74	1–5
4. Depressive complaints	−0.29 ^a^	0.26 ^a^	−0.18 ^a^	0.85	3.83	584	3.84	0–27

Note. ^a^ *p* < 0.001; ^b^ *p* = 0.01; SD = standard deviation; ranges as reported are the minimum value and maximum value possible on the scale; Cronbach’s alphas are presented on the diagonal of the table; *n* = number of complete responses to the full measure.

**Table 2 ijerph-18-04701-t002:** Stepwise estimated effects from the moderated mediation model.

	**β**	**SE(HC4)**	**t-Value**	***p*-Value**	**95% CI**
Step 1: Effect of personal resilience on worries about infections					
Personal resilience	−0.42 ^b^	0.04	−9.34	<0.001	−0.50–−0.33
Model R^2^ = 0.12 ^b^, *p* < 0.001, df(1) = 1, df(2) = 570					
	**β**	**SE(HC4)**	**t-Value**	***p*-Value**	**95% CI**
Step 2: Concurrently estimated effects of predictor variables on depressive complaints					
Personal resilience	−0.99 ^b^	0.23	−4.23	<0.001	−1.45–−0.53
Worries about infections	0.75 ^b^	0.22	3.35	<0.001	0.31–1.19
Team social climate	−0.52 ^a^	0.26	−2.03	0.04	−1.02–−0.02
Interaction Worries * Climate	−0.88 ^a^	0.40	−2.18	0.03	−1.68–−0.09
Model R^2^ = 0.14 ^b^, *p* < 0.001, df(1) = 4, df(2) = 567; R^2^-change= 0.02 ^a^, *p* = 0.03, df(1) = 1, df(2) = 567					
	**Index**	**SE(Boot)**			**95% CI (Boot)**
Step 3: Test of the full moderated mediation effect					
Index of moderated mediation	0.37 ^c^	0.16			0.07–0.73

Note. ^a^ significant at *p* < 0.05, ^b^ significant at *p* < 0.001; ^c^ significant based on bootstrapped confidence interval; SE = standard error; 95% CI = 95% confidence interval; HC4 = Crebrari-Neto correction for heteroscedasticity; (Boot) = bootstrapped estimate; the full model was estimated using IBM ^®^ SPSS ^®^ Statistics with PROCESS 3.4 plugin, model 14.

## Data Availability

Data for this study can be requested from the corresponding author.

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
