# Peer review of "We’re Not Gonna Fall: Depressive Complaints, Personal Resilience, Team Social Climate, and Worries about Infections among Hospital Workers during a Pandemic"

_ijerph, 2021, doi:10.3390/ijerph18094701_

Round 1

Reviewer 1 Report

The authors of this MS address a highly important, timely, and relevant topic, that is, the impact of the COVID-19 pandemic on health workers well-being. Overall the MS is well written. Despite my enthusiasm about this work, I have identified several major and minor issues, which will be outlined in turn.

Major issue 1.

The existing literature has comprehensively shown that well-being is a multi-facetted construct. For example, well-being has been defined in terms of a tripartite model consisting of high positive affect, low negative affect, and satisfaction with life (e.g., Diener, 2009a, 2009b; Diener, Lucas, & Oishi, 2018), or Ryff (1989) argued that well-being consists of six dimensions: self-acceptance,  positive  relations  with  others,  autonomy,  environmental  mastery, purpose in life, and personal growth. The readers is puzzled by the authors decision to operationalize well-being only via “depressive complaints”. Therefore, it remains unclear whether the rationale for the hypotheses is based on the authors’ narrow definition of well-being, or a broader definition of well-being, because previous research showed that, depending on the definition of the constructs under scrutiny, relation can lead to opposing predictions (e.g., Hackney & Sanders, 2003).

Major issue 2.

The authors rational for the proposed moderated mediation remains unclear. I highly recommend reading Fiedler, Harris, and Schott (2018) on this issue. Moreover, to provide a stronger test for the proposed direction of the effect, the authors should run all possible combination of the model to check whether their model provides the best model-fit (e.g., Fiedler et. al., 2018; Judd, Yzerbyt, & Muller, 2014).

Major issue 3.

I highly encourage the authors to include occupational group as a random factor in their model, because otherwise important information will be lost (e.g. Bates, 2010). Moreover, I would expect substantial variance that will be explained by occupational group.

Major issue 4.

Given that open science is highly important to increase reproducibility, and replicability of scientific findings, I highly encourage the authors to publish their materials, pseudonymized data, and analyses scripts on a repository such as OSF.

Minor issue 1.

The figures’ readability has to be improved.

Minor issue 2.

I highly encourage the authors to show how much direct effect of the IV on the DV reduces, after including the mediator. 

Minor issue 3.

Given that I would expect differences between full-time and part-time employees, I suggest including working hours a week as fixed factor in the model.

Author Response

Reviewer 1 Comments

Reviewer 1, Comment 1: The authors of this MS address a highly important, timely, and relevant topic, that is, the impact of the COVID-19 pandemic on health workers well-being. Overall the MS is well written. Despite my enthusiasm about this work, I have identified several major and minor issues, which will be outlined in turn.

Reply to Reviewer 1, Comment 1: We thank Reviewer 1 for their time and efforts in reviewing our manuscript and appreciate Reviewer 1's compliments on the paper in general and their insightful comments regarding specific issues with our paper. We address each of Reviewer 1's concerns below. 

Reviewer 1, Comment 2 (major issue 1): The existing literature has comprehensively shown that well-being is a multi-facetted construct. For example, well-being has been defined in terms of a tripartite model consisting of high positive affect, low negative affect, and satisfaction with life (e.g., Diener, 2009a, 2009b; Diener, Lucas, & Oishi, 2018), or Ryff (1989) argued that well-being consists of six dimensions: self-acceptance,  positive  relations  with  others,  autonomy,  environmental  mastery, purpose in life, and personal growth. The readers is puzzled by the authors decision to operationalize well-being only via “depressive complaints”. Therefore, it remains unclear whether the rationale for the hypotheses is based on the authors’ narrow definition of well-being, or a broader definition of well-being, because previous research showed that, depending on the definition of the constructs under scrutiny, relation can lead to opposing predictions (e.g., Hackney & Sanders, 2003).

Reply to Reviewer 1, Comment 2: We thank Reviewer 1 for pointing us to the richness of the well-being concept as used in other papers. We are aware of these conceptualizations, but considered depressive complaints as a feasible operationalization of well-being, as those with more depressive complaints could be seen as having lower well-being. We think it is an important point that Reviewer 1 makes here, and therefore have chosen to address it thoroughly. That is, although it is arguably debatable what exactly well-being is (i.e., given the formative approaches, well-being as a social construct, and dependency on definitions), we recognize that depressive complaints might be too narrow to be used synonymously with the term well-being or as a comprehensive operationalization of this concept. As such, we now label our main concept of interest exactly as what it is: Depressive complaints. We do now sometimes refer to it as psychological health (because evidently, having depressive complaints is indicative of lower psychological health warranting use of this term on a higher level of abstraction (e.g., Headey et al., 1993; Kato, 2012)) which is less specific than well-being. Moreover, we believe depressive complaints as a phenomenon is sufficiently interesting to warrant a study on its own and, as such, to avoid confusing our reader, we no longer use the well-being frame in the current version of the manuscript. We have made these adjustments throughout the manuscript, largely everywhere where the term ‘well-being’ occurred.

Reviewer 1, Comment 3 (major issue 2): The authors rational for the proposed moderated mediation remains unclear. I highly recommend reading Fiedler, Harris, and Schott (2018) on this issue. Moreover, to provide a stronger test for the proposed direction of the effect, the authors should run all possible combination of the model to check whether their model provides the best model-fit (e.g., Fiedler et. al., 2018; Judd, Yzerbyt, & Muller, 2014).

Reply to Reviewer 1, Comment 3: We appreciate Reviewer 1’s insightful comments on the importance of the theoretical rationale for the proposed moderated mediation model. We have included a passage in our theoretical background section to discuss this more thoroughly (lines 151-167). Specifically, we argue that since personal resilience can be considered a trait, it should be at the first step (i.e., the basal predictor) in our model (i.e., there are papers that include traits as mediator, but we find this conceptually odd in non-experimental non-longitudinal papers). Worries about infections then makes sense as a mediator variable, because it is much more momentary (i.e., it only makes sense during the pandemic). It can be seen as a specific momentary utterance of the resilience trait, as triggered by the pandemic. Depressive complaints only makes theoretical sense as outcome variable in our model, particularly because it was measured at a slightly later point in time than the predictor and mediator variables. Moreover, it is this variable we are interested in predicting based on personal resilience and worries about infections, and we are not interested in predicting the reversed relationships, nor would this be appropriate given our data. Finally, the moderator of team social climate also does not make sense in a different role in this model. That is, it is not likely to predict this variable on the basis of personal resilience and worries (because it there is no clear theoretical rationale for doing so), while it works perfectly as a resource (and has been considered as such in previous studies as well) that might buffer negative effects (in alignment with COR theory).

In our opinion, the only debate one can have is which relationship team social climate should exactly moderate in our model. For us, the relationship we have primarily targeted is of greatest interest. However, in appreciation of Reviewer 1’s comment on testing alternative models, we have ran models where team social climate moderated different relationships. First, we have estimated the effect of team social climate on the relationship between personal resilience and depressive complaints. This effect was significant (although a relatively higher p-value was found) and the R-square associated with this effect was smaller than for our original model. This makes sense, because the real moderation of interest would rather target the specific utterance of personal resilience, rather than its general effect (i.e., as we tested originally and keep in the paper). Second, we have tested this moderation effect (i.e., team social climate’s effect on the relationship between personal resilience and depressive complaints) in the presence of our initial moderation effect (i.e., team social climate’s effect on the relationship between worries about infections and depressive complaints). Here, the interaction effect between personal resilience and team social climate is no longer significant, again point to the superiority of our initially tested effect. Third, we have looked at team social climate as moderating the effect of personal resilience on worries about infections, but this was also non-significant. As such, these additional analyses suggest that our model as proposed is the most feasible. We do not include these analyses completely in our paper to avoid disturbing the flow of the paper, but have included a note in our methods (lines 264-269) and results (lines 332-343) sections that these analyses have been performed and that the output can be requested from the corresponding author (lines . However, if the Editor would like us to include all these details more extensively in the paper, we are open to doing so.

We agree completely with Reviewer 1 and Fiedler et al. (2018) on the importance of being careful in claims about causality on the basis of (observational) mediation analyses. After having carefully re-read the suggested Fiedler et al. paper (we were certainly aware of this paper, but appreciate Reviewer 1 for pointing us to this seminal work again greatly), we have scanned our paper for any framing of the identified relationships as causal. Subsequently, we have corrected the framing of relationships as correlational where appropriate.

Reviewer 1, Comment 4 (major issue 3): I highly encourage the authors to include occupational group as a random factor in their model, because otherwise important information will be lost (e.g. Bates, 2010). Moreover, I would expect substantial variance that will be explained by occupational group.

Reply to Reviewer 1, Comment 4: We thank Reviewer 1 for this methodologically relevant comment. In appreciation of this comment, we have performed several additional analyses to explore the extent to which occupational group is relevant to our main research model. First, we have performed ANOVAs with each of the variables in our model as dependent variable and occupational group as independent variable. These ANOVAs suffered from unequal group sizes threatening there robustness, but revealed no meaningful differences between the occupational groups. Second, we have performed Multilevel Regression Analyses where we nested all participants in their occupational group. We have used a stepwise approach here to first check for significant intercept variation between groups for all of the variables in our model. We did not find any significant intercept variation and intraclass correlations were below 0.05 (and within group variance exceeded between group variance tremendously), suggesting that nesting in occupational group did not account for meaningful amounts of variation in intercepts. Finally, we have estimated all of the paths in our moderated mediation model in Multilevel Regression Analyses where participants were nested in occupational groups as well, to see if there was any evidence for variation in regression slopes as a function of occupational group. Here too, we did not find any evidence for significant random effects. Taken together, it seems that nesting in occupational group did not add any meaningful explanation of variation in our models and thus we have decided to leave it out of the paper. Following these considerations and results, we have included a note on having performed these additional analyses in the methods (lines 264-269) and results (lines 332-343) sections of our paper, but do not think it adds to the narrative structure of the paper to go into further detail. However, if the Editor would require us to include this in the paper, we are open to doing so.

Reviewer 1, Comment 5 (major issue 4): Given that open science is highly important to increase reproducibility, and replicability of scientific findings, I highly encourage the authors to publish their materials, pseudonymized data, and analyses scripts on a repository such as OSF.

Reply to Reviewer 1, Comment 5: We agree that open science is highly important and are required (i.e., by the funding organization) to when our data collection has finished provide access to our data following FAIR guidelines for data-storage. However, at this point we cannot (yet) provide access to our data, because the agreement surrounding privacy does not allow us to do this, particularly because the study is still ongoing. However, we will store the meta-data of our research in Dataverse and can at a later point in time, with the hospitals' permission perhaps give full access to our data.  As noted in our manuscript, the data used for this manuscript can be requested from the corresponding author.

Reviewer 1, Comment 6 (minor issue 1): The figures’ readability has to be improved.

Reply to Reviewer 1, Comment 6: We greatly appreciate that Reviewer 1 points us towards this readability issue as we had not realized this ourselves. We completely agree and have increased the size of the figures and thereby improve their readability. However, if the Editor thinks that any specific further readability improvements are necessary, we are happy to make further adjustments.

Reviewer 1, Comment 7 (minor issue 2): I highly encourage the authors to show how much direct effect of the IV on the DV reduces, after including the mediator. 

Reply to Reviewer 1, Comment 7: We appreciate Reviewer 1’s suggestion on including details about partitioning the total effect with the mediator included. We have done this by estimating a separate mediation model (i.e., without the moderation from team social climate) and comparing the effect sizes. We have announced these additional analyses in our methods section (lines 264-269) and included the effect sizes in the results section as a post-hoc analysis in a separate paragraph (lines 315-324).

Reviewer 1, Comment 8 (minor issue 3): Given that I would expect differences between full-time and part-time employees, I suggest including working hours a week as fixed factor in the model.

Reply to Reviewer 1, Comment 8: We have estimated our entire moderated mediation model while controlling for working hours per week. Working hours per week did not have a significant effect and including it did not change any of the other estimates. We have mentioned this as one of the additional post-hoc analyses in our methods (lines 264-269) and result (lines 332-343) sections, but do not report the results more extensively to avoid disrupting the narrative flow of the paper. That is, any variable in a model would require a theoretical rationale and we do not have a strong theoretical basis to expect a (confounding) effect of working hours in this specific context. Therefore, including it as a sole confounder in the model would be a bit random. Instead, the results from this analysis are positioned as being available upon request from the corresponding author. However, if the Editor would like us to include this post-hoc analysis in the paper in more detail, we are certainly open to doing so.

References used:

Headey, B., Kelley, J., & Wearing, A. (1993). Dimensions of mental health: Life satisfaction, positive affect, anxiety and depression. Social Indicators Research, 29(1), 63-82. https://doi.org/10.1007/bf01136197

Kato, T. (2012). Development of the Coping Flexibility Scale: Evidence for the coping flexibility hypothesis. Journal of Counseling Psychology, 59(2), 262-273. https://doi.org/10.1037/a0027770

Reviewer 2 Report

This study aims to explore the relationships among personal resilience, team social climate and wellbeing utilising Conservation of Resources (COR) theory. Using data from 1126 healthcare workers in the Netherlands found that:

-resilience negatively relates to depressive complaints

- resilience reduces worries about infections

- Worries about infections are positively related to depressive complaints

- resilience is an important resource in maintaining wellbeing during a pandemic.

Line 130 – offer not offers

Were there any measures put in place if the participant answered yes to final question of the PHQ9?

Why did healthcare workers choose not to participate in the study?

Author Response

Reviewer 2 comments:

Reviewer 2, Comment 1: This study aims to explore the relationships among personal resilience, team social climate and wellbeing utilising Conservation of Resources (COR) theory. Using data from 1126 healthcare workers in the Netherlands found that: resilience negatively relates to depressive complaints; resilience reduces worries about infections; worries about infections are positively related to depressive complaints; resilience is an important resource in maintaining wellbeing during a pandemic.

Reply to Reviewer 2, Comment 1: We thank Reviewer 2 for their time and efforts reviewing the manuscript and their positive evaluation. The descriptive comments Reviewer 2 provides here form a great description of our paper.

Reviewer 2, Comment 2: Line 130 – offer not offers

Reply to Reviewer 2, Comment 2: Thank you very much, we have adjusted it.

Reviewer 2, Comment 3: Were there any measures put in place if the participant answered yes to final question of the PHQ9?

Reply to Reviewer 2, Comment 3: Thank you for pointing out this important consideration, particularly in the context of doing the research. To answer your question: Yes, in so far that we included reference to professional care with those questions. The hospitals participating in our sample had psychosocial response teams that could be contacted, and we included the national helpline's number for people answering the questions. However, given privacy considerations and research ethics we could not act on individual participants' responses to this item. We have included a sentence in our methods section indicating the presence of connections to these help-services in the survey.

Reviewer 2, Comment 4: Why did healthcare workers choose not to participate in the study?

Reply to Reviewer 2, Comment 4: We thank Reviewer 2 for this question. Since we sampled from a large amount of employees, it can be expected that not all of them participate. It is unclear to us what exactly the reasons for non-participation are, because non-participants constitute a very large and diverse group. We did not purposefully exclude people from participation in any way, so such reasons do not apply. We have added a sentence on this in the limitations section of our paper.